# The effect of face masks on confusion of emotional expressions

**Andrea R. Libby[◔], Collin Scarince[iD][◔]\***

Department of Psychology & Sociology, Texas A&M University – Corpus Christi, Corpus Christi, Texas, United States of America

◔ These authors contributed equally to this work.
\* Collin.Scarince@tamucc.edu

## Abstract

While the use of personal protective equipment protects healthcare workers against transmissible disease, it also obscures the lower facial regions that are vital for transmitting emotion signals. Previous studies have found that face coverings can impair recognition of emotional expressions, particularly those that rely on signals from the lower regions of the face, such as disgust. Recent research on the individual differences that may influence expression recognition, such as emotional intelligence, has shown mixed results. In the current investigation, we conducted two experiments to build on previous work to further investigate the role of emotional intelligence in emotion recognition. Participants viewed a set of masked and unmasked models displaying six different emotional expressions for a limited time (0.5 s, 1.0 s, or 1.5 s in Experiment 1 and 1.0 s in Experiment 2). Participants then indicated which emotion options they felt each model expressed. Participants' emotional intelligence was measured using the Schutte Self Report Emotional Test. Emotion recognition accuracy was lower for images with masks than images without masks in both experiments. Confusion rates were not uniform: masked disgust and sadness had the greatest increase of confusion from their unmasked conditions, happiness and surprise had the least confusion in all conditions, and both fear and anger had generally low recognition in masked and unmasked conditions. Emotional intelligence only impacted accuracy when view time was particularly restricted and parts of the face were covered. These results replicate the finding that face masks impair the transmission of emotional signals, and the impairments are a function of the occluded facial regions. Emotional intelligence may moderate this effect in dynamic social interactions where emotional expressions are brief and ambiguous.

**Data availability statement:** Data from the two experiments reported here are available via the Open Science Framework repository for this project at https://osf.io/wrdyv/.

**Funding:** The author(s) received no specific funding for this work.

**Competing interests:** The authors have declared that no competing interests exist.

## Introduction

The use of personal protective equipment, such as face masks, protects healthcare workers and others against transmissible diseases, but these masks also obscure facial features that are vital for interpreting emotion expressions. During and following the COVID-19 pandemic, researchers studied how facial coverings influence perception of emotional states (see Barnett [1] and Ramdani et al. [2] for reviews) and how those interpretations may affect social interactions [3–5]. Building on this previous work, the current research investigated how facial coverings affect emotion recognition accuracy and the rate at which certain facial expressions are confused with others while also accounting for individual differences in processing emotional information.

### Specialization of Emotional Expressions

The Basic Emotion Theory [6] proposes that nonverbal expressions of emotion evolved as automatic outwardly responses of internal emotional states. Evidence suggests there are seven universal basic emotions: anger, fear, surprise, sadness, disgust, contempt, and happiness [7,8] (although others have proposed as many as 24 distinct expressions [6]). The distinctiveness of the basic expressions is a result of the activation of different facial muscles, or action units (AUs). For example, the expression for happiness (the Duchenne smile) is characterized by raised cheeks (AU 6) and upward-pulled lip corners (AU 12) [9]. Due to their innate nature, these signals are interpreted quickly and do not always require conscious perception to detect or express [10–13].

Some emotional expressions rely more on AUs in certain areas of the face than others [14–16]. Differentiation of emotions occurs within AUs across the face, although there remains some disagreement about which emotions rely more heavily on each region [17,18]. For instance, while surprise and happiness can both be identified by an open mouth, happiness can be differentiated from surprise within this region based on raised lip corners and teeth visibility [15,19] and, outside of this region, surprise can be differentiated by raised eyebrows. Other basic emotions, like anger and fear, rely more on AUs in the eye regions, such as raised upper eyelids or eyebrows [19].

Given that distinct regions of the face are important for interpreting facial expressions, it is not surprising that expression recognition may be impaired if certain parts of the face are not visible [20–25]. While both the eye region and mouth region are significant informants of emotions, Kotsia et al. [17] and Blais et al. [16] determined that occluding the mouth region impairs overall emotion recognition more than occluding other facial regions. Accuracy is further reduced for expressions that have more key AUs in the mouth region than other facial regions, such as disgust [18,26,27]. These findings together indicate that limited visibility of the lower half of the face, such as when a face mask is worn, will impair overall emotion recognition by blocking the observer from necessary diagnostic information. These findings also indicate that face coverings will impair recognition of some emotional expressions more than others, based on the diagnostic information necessary for identification of that expression [27].

## Emotion recognition and face masks

Several studies investigated the effects of face makes on interpreting facial expressions in response to the COVID-19 pandemic. Many of these studies involve participants viewing images or videos of a person (target) modeling a basic emotion either with a face covering or not. In line with pre-pandemic research, recent studies have found face masks on targets reduce facial expression recognition accuracy [28–34], dampen perceived intensity of emotion [35,36], and lower confidence in the interpretation of emotions [29–31,37]. In addition, face masks impact recognition differentially based on the AUs covered by face masks. For example, emotions that are expressed more with the eyes, such as anger, are less affected by face masks than other expressions [38]. Research on broader effects on social perception has found wearing facial coverings lowers feelings of trustworthiness toward the target [33,39] and reduces feelings of interpersonal closeness [39,40].

Carbon [29] published one of the earliest studies that specifically investigated how face masks influence emotion recognition in response to the mass adoption of facemasks. All participants viewed a set of images fully crossed on sex (male, female), age (young, middle-aged, elderly), emotion (angry, disgusted, fearful, happy, neutral, sad), and visibility (mask, no mask). Participants viewed each image without time limit, indicated which emotion they thought was expressed by the target, then rated their response confidence. Accuracy of emotion recognition and the confidence in their responses declined when evaluating masked stimuli compared to unmasked stimuli for angry, disgusted, happy, and sad expressions. Carbon also found that angry, disgusted, happy, and sad expressions were all consistently misidentified as neutral. Accuracy did not decline in the case of masked fearful or neutral expressions, which Carbon attributed to ceiling performance, though participant response confidence did decline.

Post-pandemic research on emotion recognition has also investigated individual differences that could predispose people to recognition errors. For example, gender has been found to affect recognition accuracy for some target expressions better than others [41,42] rather than affecting overall accuracy. Emotion recognition for covered faces also tends to decrease with age [26,39], though one study found that toddlers struggled to identify masked emotions [43]. Furthermore, long-term exposure to people wearing face masks in public is associated with better emotion recognition in children [44]. Individual differences in mental wellness have also been investigated. Men with high perceived stress have been found to take longer to identify masked happiness than men with low or moderate stress while the inverse was found for women [45].

## Emotional Intelligence

Another potential individual difference relevant to interpreting emotional expressions is emotional intelligence (EI) [46,47]. According to the four-branch ability model [48,49], EI is a processing ability composed of four skill families including perceiving emotions, using emotions to facilitate thought, understanding emotions, and regulating emotions [50,51]. Individuals with high EI tend to recognize emotional expressions faster than those with low EI [52]. This suggests that EI can likely inform how individuals interpret expressions under ambiguity, such as when wearing face coverings.

In a follow-up to earlier work [29], Carbon et al. [30] investigated how EI impacted expression recognition, measuring participants' EI on both a self-reporting and ability-based level. Participants again evaluated masked and unmasked targets and assigned one of six emotional states to each image: anger, disgust, fear, happiness, sadness, and neutral. Although the previous findings that emotion recognition accuracy decreases in masked conditions were replicated, there was no significant effect of EI, either self-reported or ability-based, on emotion recognition accuracy.

Recently, Huc et al. [53] investigated how several psychological individual differences—including perceived stress, loneliness, psychological disorders, and EI—affect expression recognition when faces are covered. Unlike Carbon's [30] procedure, targets were presented for only 450 ms. Their results indicated that EI was a positive predictor for recognition accuracy, particularly for happy expressions. The difference in target presentation duration might account for the discrepancy between Huc et al.'s positive findings and Carbon's null results. EI may only reliably predict expression recognition when there is high ambiguity. When an expression is unobstructed and visible for an extended time, both high- and low-EI

individuals can effectively process the expression. The less clear the signal is, like when part of the face is covered and the expression is fleeting, the more individual differences in processing emotional signals might affect recognition.

## Current investigation

The current study aimed to further investigate the effects of face masks on emotion perception. Specifically, we wanted to extend Carbon's [29,30] findings to instances where there is limited time to process a facial expression [53]. Due to the rapid processing of facial expressions [10,13,40,54,55], the influence of EI could be mitigated when participants have unlimited time to review the target. To that end, we restricted how long targets were visible before participants responded in our experiments. We hypothesize that (1) emotion recognition accuracy will be lower when images are masked as opposed to unmasked images, (2) that emotion recognition will decline as observation time will decline, and (3) EI would have a positive relationship with the ability to accurately identify emotional expressions.

## Experiment 1

### Method

**Participants.** We recruited 105 participants through Prolific (https://www.prolific.co/) for Experiment 1. Of the 105 respondents collected, 100 were retained for data analysis after checking for completeness and accuracy. The sample was balanced between men and women (52% women, 47% men, 1% non-binary/other gender), and ages ranged between 20 and 69 years old ($M = 38.46$, $SD = 12.71$). Participants were predominately White (70% White, 9% Black/African American, 9% Latino/Hispanic, 8% Asian, 4% other/missing) and were all citizens of the United States. Inclusion criteria also required participants to take the survey on a desktop or laptop computer. Recruitment began and concluded on 5 May 2022. Participants were compensated $2.50 upon completion of the survey.

### Ethics statement

All procedures used in Experiment 1 and Experiment 2 were approved by the Institutional Review Board at Texas A&M University – Corpus Christi and classified as *exempt* for benign behavioral intervention (TAMU-CC-IRB-2022–0432). All participants reviewed a written electronic consent form online and provided electronic consent to participate before beginning the procedure.

### Materials

**Emotion Stimuli.** All image stimuli were collected from the Extended Cohn Kanade Dataset (CK+) [56,57]. The CK + is a dataset containing photos of human targets displaying different emotional expressions specified by, and fit to, AUs. Forty-two images were selected as targets (23 female, 19 male), displaying six different emotional expressions: happiness, anger, surprise, fear, disgust, and sadness. Neutral expressions were excluded as the photos available were not easily crossed with other stimuli variables, and to maintain an appropriate median completion time. No model appeared in the survey more than three times, and no model displaying the same emotion was used in both the masked and unmasked condition. The experimental stimuli consisted of 31 images of male targets (15 masked, 15 unmasked, and 1 unmasked example) and 31 images of female targets (15 masked, 15 unmasked, and 1 masked image used in an attention check question). The stimuli consisted of an equal number of each emotional expression between masked and unmasked conditions: 6 happiness, 5 anger, 6 surprise, 5 fear, 3 disgust, and 6 sadness. Images were partially crossed on levels of age (old, young) race (white, not white), and sex (male, female) for both masked and unmasked images.

We digitally superimposed an image of a standard surgical mask onto half the dataset. All images that were not originally black and white were desaturated to maintain consistency across stimuli. An example of an unmasked and masked target can be found on the OSF repository for this study (https://osf.io/5f84x).

**Emotional Intelligence.** We measured EI using the Schutte Self Report Emotional Test (SSEIT) [58], similar to Huc et al. [53]. This test consists of 33 items (α = .935) representing Salovey and Mayer's [48] conceptualization of EI, each a short statement to which participants rate their agreement to on a 5-point Likert scale ranging from "strongly disagree" to "strongly agree".

### Design and procedure

We implemented a 2 (mask: masked, unmasked) × 6 (emotion: happiness, anger, surprise, fear, disgust, sadness) × 3 (timing: 0.5 s, 1.0 s, 1.5 s) mixed design with mask and emotion as within-subjects factors, observation time as a between-subjects factor, and expression recognition accuracy as the dependent variable. Participants' EI scores (measured via the SSEIT) were used as a continuous factor.

This study was conducted online via Qualtrics (https://www.qualtrics.com/). Participants first read an instructional page describing the process of viewing and assessing the images, which included an example image. Next, the participants completed the image observation trials. Each trial consisted of viewing a single expression image for either 0.5 s, 1.0 s, or 1.5 s. When the time was complete, the image was immediately replaced by a multiple-choice question prompting participants to indicate what emotion was expressed in the previous image. This question included ten response options including six basic emotions (anger, fear, surprise, sadness, disgust, and happiness) as well as four lures that were never the correct response (neutral, tiredness, annoyance, and excitement) [39]. Participants used their cursor to make a response. All participants viewed the same set of 60 masked or unmasked images displaying six emotional expressions. After completing the image observation trials, participants completed the SSEIT followed by demographic questions.

### Analysis strategy

Responses were first evaluated for completeness and accuracy prior to analysis. Four participants were removed for not completing the full procedure. Of the remaining respondents, one was removed for not answering three questions on the SSEIT. To avoid emotion recognition accuracy outliers, we set a criterion of 2.5 standard deviations below the condition mean. One additional participant was removed for scoring below this threshold. Due to the experiment being conducted online outside of a controlled setting with standardized equipment and the number of possible responses, we did not analyze response times. All analyses were run using IBM SPSS (v29), and we set a significance criterion of α = .05 for all tests. All post hoc tests used Bonferroni-corrected confidence intervals for pairwise comparisons.

We had two goals when analyzing responses: (1) analyze the influence of mask, timing, and EI on emotion recognition accuracy, and (2) identify which expressions were highly confusable with each other. For our first goal, we planned to analyze the effects of mask and timing on accuracy using a 2 (mask) × 6 (emotion) × 3 (timing) ANCOVA with EI as a covariate. We used a linear model with the categorical independent variables and EI as factors to check the assumption of homogeneity of regression slopes and found a significant EI by timing interaction, $F(2, 99) = 4.37$, $p = .015$, which violated this assumption. We subsequently adjusted our statistical approach. Instead, we adopted a two-step strategy to first analyze the effects of the categorical variables using a 2 × 6 × 3 ANOVA, and second to investigate the timing by EI interaction using linear regression with mask, timing, and the mask × timing interaction as predictors. For our second goal, we calculated the frequency of each of the ten possible responses separated by factors of mask and target emotion. We used these matrices to identify patterns of confusability for incorrect responses [29,30]. We used the results of the ANOVA to test the effects of masks on general recognition of particular emotions.

## Results and discussion

### Accuracy

We first used a 2 (mask) × 6 (emotion) × 3 (timing) ANOVA to analyze the categorical effects on accuracy (Greenhouse-Geisser corrected degrees of freedom used for effects that included the emotion factor for violations of sphericity). Masks had a significant main effect on recognition accuracy, $F(1, 97) = 382.31$, $p < .001$, $\eta_p^2 = 0.798$. Performance declined from a

total mean accuracy of 76.67% ($SD=9.48\%$) in the no mask condition to a total mean accuracy of 50.34% ($SD=10.99\%$) in the mask condition. There was also a significant main effect of emotion, $F(3.33, 323.40) = 87.61$, $p<.001$, $\eta_p^2=.475$. Happiness ($M=85.0\%$, $SD=18.2\%$) and surprise ($M=82.9\%$, $SD=20.5\%$) resulted in higher recognition rates than the other emotions. Fear ($M=52.6\%$, $SD=28.6\%$), disgust ($M=63.1\%$, $SD=34.3\%$) and sadness ($M=53.7\%$, $SD=41.6\%$) had higher recognition than anger ($M=42.9\%$, $SD=23.6\%$). There was no significant main effect of timing, $F(2, 97) = 0.50$, $p=.606$.

The main effects of mask and emotion were qualified by a significant two-way interaction, $F(4.25, 412.12) = 101.62$, $p<.001$, $\eta_p^2=.512$ (see Fig 1). Accuracy in the masked conditions were lower for all emotions except for fear ($p=.099$). The effect was particularly pronounced for sadness. There was not a significant interaction between mask and timing, $F(2, 97) = 0.01$, $p=.995$, nor was there a significant three-way interaction, $F(8.50, 412.12) = 0.76$, $p=.648$.

Mean accuracy rate for each emotion by mask condition. Lines labeled with letters indicate which pairs of target emotions resulted in equivalent accuracy. Significant simple effects between mask conditions at each level of target emotion are noted with asterisks (*). Error bars represent one standard error around the mean.

Multiple regression analyses were then used to investigate the interaction between the timing conditions and EI. Separate regressions were conducted for each level of the mask condition. Centered EI scores, timing, and the EI×timing interaction were entered in the same step of each model. The results of the regression analyses are presented in Table 1. The overall model in the mask condition was near significance, $F(3, 96) = 2.58$, $p=.058$, $R^2=.74$. The interaction term was significant, $t(96) = 2.11$, $p=.037$, $\beta=0.22$, but the main effects of timing and EI were not significant ($p$'s>.05). This interaction indicates that EI was a positive predictor of accuracy when time was limited (simple slopes in the 0.5s condition $\beta=0.32$ and 1s condition $\beta=0.32$), but the effect was weaker and negative at longer view times (simple slope in the 1.5s condition $\beta=-0.23$). For the condition in which the targets did not have a mask, the overall model was not significant, $F(3, 96) = 1.12$, $p=.345$, $R^2=.03$.

Coefficients from the linear regression to analyze the interaction between EI and time viewing an expression between mask conditions.

## Confusion patterns

To further investigate these effects on emotion recognition, we composed a set of confusion matrixes of the total percent of responses given for each target emotion (Fig 2). We organized these erroneous responses into the no mask and mask conditions to investigate the patterns between conditions.

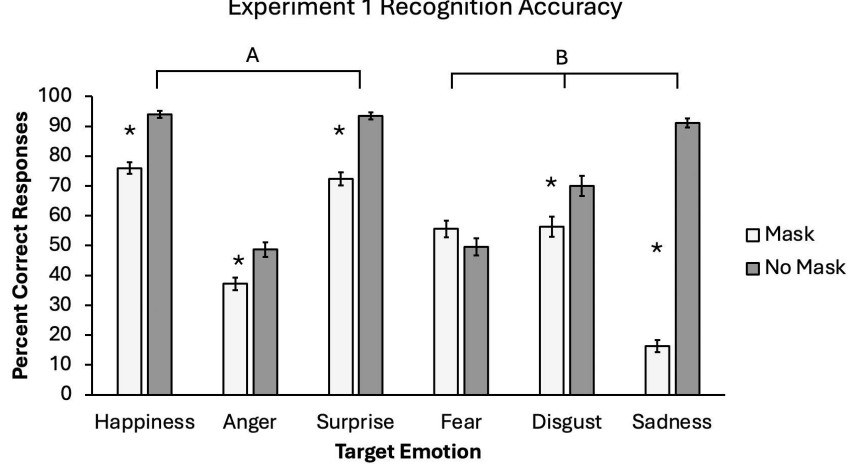

**Fig 1. Emotion recognition accuracy for Experiment 1.**

**Table 1. Linear regression coefficients.**

| Variables | B | β | SE | t | p | 95% CI |
|---|---|---|---|---|---|---|
| | | | **Mask** | | | |
| Constant | 50.31 | | 1.10 | 46.29 | <.001 | [48.00, 52.30] |
| EI | 0.08 | 0.12 | 0.06 | 1.17 | .245 | [-0.05, 0.20] |
| Time | 0.10 | 0.004 | 2.67 | 0.04 | .970 | [-5.13, 5.32] |
| EI × Time | 0.33 | 0.22 | 0.002 | 2.11 | .037 | [0.02, 0.65] |
| | | | **No Mask** | | | |
| Constant | 76.55 | | 0.95 | 80.85 | <.001 | [74.67, 78.43] |
| EI | 0.05 | 0.10 | 0.06 | 0.94 | .351 | [-0.06, 0.16] |
| Time | −0.56 | −0.03 | 2.30 | −0.25 | .807 | [-5.13, 4.00] |
| EI × Time | 0.17 | 0.13 | 0.14 | 1.25 | .213 | [-0.10, 0.45] |

**Fig 2. Experiment 1 expression confusion rates across mask conditions.**

This confusion matrix compares the total rate of responses given for each target expression in the mask and no mask conditions in Experiment 1. The emotion displayed in the image (target) are arranged along the columns. Rate of responses are arranged along the rows, the last four of which were only used as lure responses. Darker cell shades correspond to higher response rates. Error responses are highlighted in red and correct responses are highlighted in green.

With the exception of fear, confusion rates increased in the mask condition compared to the no mask condition. Neutral and excitement comprised a majority of erroneous responses for both happiness and surprise in the mask condition. Confusion of disgust in the mask condition was the most diverse compared to responses in the no mask condition, in which disgust was only confused with either annoyance or anger. Overall accuracy of identifying anger was less than 50%, with the primary erroneous response being annoyance. Of the lure responses, annoyance was frequently selected in place of anger, disgust, and sadness.

In Experiment 1, we investigated how viewing emotions with restricted time would affect emotion expression recognition and the role of EI as a predictor in performance. While we did not find a direct effect of timing, our results generally confirm previous findings on emotion confusion patterns [29,30,39]. The performance decline from a total mean accuracy is consistent with the findings by Grundmann et al. [39], who found that emotion recognition decreased from 69.9% in the no mask condition to 48.9% in the mask condition when including additional lure responses (e.g., proud, amused, etc.). As with Carbon [29,30], fear was the exception to this trend, wherein accuracy did not decrease when images were masked as opposed to unmasked.

Contrary to our hypothesis, EI was not strongly associated with accuracy. Although EI did not affect accuracy alone, the interaction between EI and view time suggests those with high EI performed better than those who scored low in EI when they were allowed only a short amount of time to view the masked faces. As view time increased, however, the benefit of high EI on recognition accuracy decreased, indicating that higher EI is only beneficial towards emotion recognition in very limited time frames.

The error rates we found in this experiment were comparatively higher than those reported in previous studies [29,30,39]. This discrepancy may be due to the inclusion of the additional response options, because lure options were consistently confused with target emotions in some conditions. For example, annoyance was frequently selected for masked anger, disgust, and sadness. Indeed, distractor responses accounted for between 16% (fear) and 93% (happiness) of errors across emotions in the masked condition. To replicate these effects with a simplified procedure more comparable to studies that did not use lures [29,30,32,33,41,59] we conducted a second experiment that only included basic emotions for response options while still restricting how long targets were visible to investigate the role of EI.

## Experiment 2

We conducted Experiment 2 to further investigate the confusion patterns and the possible effect of EI after addressing limitations of Experiment 1. First, we eliminated the timing condition and kept the limited viewing time consistent to focus our efforts on the effects of face coverings and EI. We selected the 1 s interval to restrict view time enough to make the recognition task difficult, but not too brief to risk presentation errors due to the experiment being conducted online. Second, we removed lure responses to the emotion recognition task, as some options in Experiment 1 could reasonably be conflated with one another based on subtle differences and interpretation of expressions. For instance, annoyance was frequently conflated with anger in both masked and unmasked conditions. Furthermore, neutral could be used as a functional "default" response when there are not strong signals for an expression given its higher response rate in the masked conditions for several target emotions. Removing these options allowed us to more accurately investigate the reliance on specific AUs for recognition, and the confusion between the target emotions.

### Method

#### Participants

We recruited 45 participants through the same crowdsourcing platform used in Experiment 1. All 45 respondents were retained for data analysis after checking for completeness and accuracy. Participants were equally mixed on the basis of gender (49% women, 47% men, 4% non-binary/third gender), and ages ranged between 18 and 71 years old ($M = 32.23$, $SD = 12.54$; one not reported). Participants were again predominately White (71% White, 9% Black/African American, 9% Latino/Hispanic, 7% Asian, 2% Middle Eastern, 2% Multiracial) and citizens of the United States. Recruitment began and concluded on 30 August 2022. Participants were compensated $2.50 upon completion of the survey.

All procedures were approved by the institution's IRB, and all participants completed an electronic written consent form before beginning the procedure.

## Materials

All image stimuli were the same between Experiments 1 and Experiment 2 except for seven images with low accuracy (3 anger, 3 sadness, 1 fear) that were replaced with new images. The sex and race of the replacement image models were consistent with the previous images. As with Experiment 1, no target appeared in the survey more than three times, and no target displaying the same emotion was used in both the masked and unmasked condition. We again measured participants' EI using the same measure from Experiment 1.

## Design and procedure

We implemented a 2 (mask) × 6 (emotion) within-subjects design with emotion recognition as the dependent variable and EI scores as a continuous measure. The procedure was the same as Experiment 1 except (1) all images were displayed for 1 s and (2) the response options (anger, fear, surprise, sadness, disgust, and happiness) did not include the four lure responses in Experiment 1.

## Analysis strategy

Responses were first evaluated for completeness and accuracy prior to analysis. No participants were removed for not completing the full procedure or for having an average accuracy 2.5 standard deviations below the sample average. Similar to our analysis approach in Experiment 1, we aimed to analyze the influence of mask-wearing and EI on recognition using a 2 (mask) × 6 (emotion) ANCOVA with EI as a covariate (there were no violations of the homogeneity of slopes across conditions in this case) and identify which basic emotions were highly confusable with each other using confusion matrices.

## Results and discussion

### Accuracy

We analyzed accuracy with an ANCOVA with mask as a within-subjects factor and EI as a covariate. EI was not a significant covariate, $F(1, 43) = 1.16$, $p = .287$, so the effect of mask and emotion on accuracy was analyzed without this covariate due to the extraneous noise it introduced to the model (Greenhouse-Geisser corrected degrees of freedom used again for effects that included the emotion factor for violating the assumption of sphericity). Consistent with the results of Experiment 1 and with the hypothesis that emotion recognition accuracy will be lower when images are masked as opposed to unmasked, masks had a significant effect on participant accuracy, $F(1, 44) = 156.21$, $p < .001$, $\eta_p^2 = .780$. Accuracy in the no mask condition was higher ($M = 83.28\%$, $SD = 8.18\%$) than the mask condition ($M = 70.11\%$, $SD = 10.00\%$). There was also a main effect of emotion, $F(3.23, 141.93) = 59.54$, $p < .001$, $\eta_p^2 = .575$. Again, happiness ($M = 96.5\%$, $SD = 9.5\%$) and surprise ($M = 92.2\%$, $SD = 12.5\%$) resulted in the highest recognition accuracy. Anger ($M = 72.0\%$, $SD = 25.4\%$) and sadness ($M = 75.4\%$, $SD = 24.3\%$) had higher accuracy than fear ($M = 55.1\%$, $SD = 24.6\%$) and disgust ($M = 51.9\%$, $SD = 37.1\%$).

Finally, there was a significant interaction between mask and emotion, $F(3.19, 140.40) = 17.24$, $p < .001$, $\eta_p^2 = .282$ (see Fig 3). There were not significant differences between the mask and no mask conditions for anger ($p = .383$) or fear ($p = .268$), but all other emotions resulted in lower accuracy in the mask conditions. In this case, disgust resulted in a considerably large difference between the mask conditions, and the effect for sadness was still substantial.

Mean accuracy rate for each emotion by mask condition in Experiment 2. Lines labeled with letters indicate which pairs of target emotions resulted in equivalent accuracy. Significant simple effects between mask conditions at each level of target emotion are noted with asterisks (*). Error bars represent one standard error around the mean.

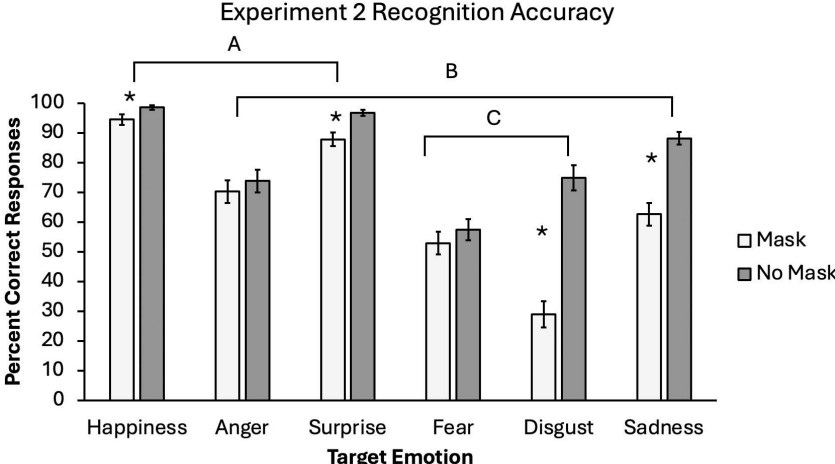

Fig 3. Emotion recognition accuracy for Experiment 2.

## Confusion patterns

As with Experiment 1, we created confusion matrices for responses in both conditions to investigate the effect masks have on confusion rates. The total response rate for each emotional expression is shown in Fig 4.

This confusion matrix compares the total rate of responses given for each target expression in the mask and no mask conditions in Experiment 2. The emotion displayed in the image (target) are arranged along the columns. Rate of responses are arranged along the rows. Darker cell shades (red for error responses and green for correct responses) correspond to higher response rates.

Rates of confusion for most emotions changed between the mask condition and no mask condition. Participant accuracy of identifying masked disgust was the lowest of all conditions, with both anger and happiness given as a response more

| Experiment 2 Response Rate (%) by Target Emotion and Mask Condition | | | | | | |
|---|---|---|---|---|---|---|
| **No Mask Targets** | | | | | | |
| | Target Emotion | | | | | |
| | Happiness | Anger | Surprise | Fear | Disgust | Sadness |
| Happiness | 98.52 | 0 | 0 | 0 | 1.48 | 0 |
| Anger | 0 | 73.78 | 0 | 0.89 | 22.22 | 2.59 |
| Surprise | 0.74 | 1.33 | 97.03 | 9.78 | 0.74 | 2.22 |
| Fear | 0.74 | 0 | 1.49 | 57.33 | 0 | 1.11 |
| Disgust | 0 | 15.11 | 0 | 28.44 | 74.81 | 5.56 |
| Sadness | 0 | 9.33 | 0.74 | 3.56 | 0.74 | 88.15 |
| **Masked Targets** | | | | | | |
| | Target Emotion | | | | | |
| | Happiness | Anger | Surprise | Fear | Disgust | Sadness |
| Happiness | 94.44 | 0 | 5.2 | 1.34 | 25.93 | 1.85 |
| Anger | 1.11 | 70.22 | 0 | 0 | 36.3 | 8.15 |
| Surprise | 2.22 | 3.11 | 88.1 | 29.46 | 5.93 | 2.59 |
| Fear | 0.04 | 8 | 1.49 | 53.13 | 0 | 10.74 |
| Disgust | 0.37 | 12.44 | 2.97 | 10.27 | 28.89 | 14.07 |
| Sadness | 1.48 | 5.78 | 2.23 | 5.36 | 2.96 | 62.59 |

(Selected Emotion is the row label for both matrices.)

Fig 4. Experiment 2 expression confusion rates across mask conditions.

often than disgust itself. Anger, fear, and sadness were all consistently confused with disgust. While masked anger and masked disgust were often confused for each other, many confusions were one directional. For example, masked disgust was often confused for happiness, but masked happiness was rarely confused for disgust; masked fear was often confused for surprise, but masked surprise was rarely confused for fear.

Consistent with Experiment 1 and previous research [29,30,39] we found face masks resulted in a significant decline in emotion recognition accuracy. Regarding confusion rate patterns between mask conditions, we found more varied error responses across most emotions in the masked condition, although anger, fear, and happiness were more robust against confusion in general. In this experiment, we did not find a relationship between EI and recognition accuracy in general or for either mask condition. Although the results of Experiment 1 suggested a relationship between accuracy and EI would be found in the masked condition, the removal of the lure responses made the recognition task easier, thus reducing the effect of EI.

## General discussion

We conducted two experiments to test the hypotheses that (1) emotion recognition accuracy will be impaired by face masks, (2) recognition will decline when there is little time to view the expression, and (3) that such errors will be less significant for people with high EI.

We found consistent evidence for our first hypothesis and found that particular expressions are more vulnerable for confusion compared to others (in line with recent research [29–31,34,36,42,60]). We did not find support for our second hypothesis, likely due to the fact that even the briefest presentation time used in this study was not extremely limited [10,55]. We found mixed evidence for our third hypothesis, in that EI only has a meaningful impact when there are several barriers to reading an expression (i.e., limited time and obscured parts of the face).

## Recognition and confusion

Face mask occlusions on targets resulted in lower overall recognition accuracy than targets without face masks in both experiments. Accuracy was comparatively higher in Experiment 2 than Experiment 1 across emotions except some cases of anger and disgust, likely due to the inclusion of the lure responses in Experiment 1. Regarding specific emotions, accuracy decreased in the masked conditions in all cases except for anger in Experiment 1 and fear in both experiments. Particularly, our findings in Experiment 2 closely align with those of Proverbio and Cerri [38] who measured the recognizability of emotions in conditions of masked and unmasked targets. They found a large effect of masks on the recognizability of disgust and sadness, no effect of masks on anger, and low overall recognizability of fear and disgust.

Patterns of confusion rates between emotions were also mostly consistent between experiments. These shifts between unmasked and masked accuracies may lie in the visible AUs defined by the Facial Action Coding System (FACS) [9,61] that describes the facial actions of happiness, surprise, anger, disgust (contempt), fear, and sadness [19,62–64].

Overall, happiness and surprise resulted in high recognition rates compared to other emotions, aligning with previous findings [31,32,35,59]. Key indicators for these two emotions—raised lower eyelids (AUs 6) for happiness and raised eyebrows and upper eyelids (AUs 1 and 5) for surprise—are left unobscured by face coverings. Masked happiness and surprise are still highly recognizable because these visible AUs around the eyes are distinct from the other expressions. Confusing happiness and surprise with a neutral expression [29,30] may be due to insufficient visible diagnostic information or lack of response confidence. The shared AUs between happiness and surprise also account for how frequently they are confused for each other [55].

Recognition of sadness and disgust were both negatively affected by face coverings, in line with previous findings [65]. One of the key indicators of sadness is lowered lip corners (AU 15), which is unique among the other basic emotions. Likewise, disgust is easily distinguishable with a raised upper lip (AUs 9 and 10) [19]. The sharp decline in accuracy in masked images of sadness and disgust is likely due to the occlusion of these AUs, highlighting their significance for identifying these

expressions. The two visible indicators of sadness, AU 1 and AU 4, are both shared with fear, leaving only AU 5, characteristic of fear, to differentiate the two, likely accounting for the confusion found between them. AU 4 is also a component of anger, which was a common error for both sadness and disgust targets.

Fear and anger had overall low recognition in both mask and unmasked conditions. Fear was most often confused with surprise, which shares many of the same AUs (AUs 1, 5, 26, and 27) [19]. Since AUs 26 and 27 are not visible on images with face masks an eyebrow frown (AU 4) is left to differentiate the expressions. The low recognition for fear is also consistent with previous studies that have found lower overall accuracy for fear [42,59] and limited effects of masks [36,42] or no effect [29–31,60]. The AUs characteristic of anger around the eyes (AUs 4, 5, and 7) are all shared with at least two other emotions [19]. This may account for the relatively low accuracy in recognizing anger. The high rates of confusion with disgust may be attributed to the shared AUs 4 and 7, while the raised upper lip (AU 9) important for identifying disgust is obscured in masked conditions. Similarly, sadness shares AU 4 with anger, one of only two visible indicators of sadness when images are masked while AU 4 and 5 are shared between anger and fear, leaving only one of three visible indicators of fear. This suggests that confusion patterns for masked anger relate to lack of information for other emotions more than for lack of identifiers of anger. The inconsistent findings for anger between the experiments reported here reflect the inconsistency from previous studies. Some investigations have found effects of masks on recognition of anger [29,31,32,42,60] while others have found no difference [40,41,59]. Indeed, Mikayzaki et al. [35] also found mixed effects between the two experiments they conducted.

### Emotional intelligence and timing

As with previous research, there was not a consistent effect of EI on expression recognition [30,53]. In Experiment 1, we found that participants with high EI were more accurate than those low in EI only when faces were covered and they viewed the images for a short amount of time. This aligns with recent findings of Huc et al., [53] that EI was positively associated with emotion recognition for masked happiness (and, to a lesser extent, recognition of masked emotions in general). In their investigation, images of masked expressions were visible for 490 ms, similar to the 500 ms in the current study. In the current experiments, EI did not have a strong relationship with accuracy when participants viewed images for a longer amount of time (1.5 s in Experiment 1 and 1 s in Experiment 2).

Our findings help reconcile the null findings reported by Carbon et al. [30] and positive findings by Huc et al. [53] in that EI was only an effective predictor of emotion recognition when time was limited and faces were covered. When there is little ambiguity in a facial expression, individual differences related to emotional intelligence play a small role in predicting accuracy because recognition of a clear signal is not difficult. In instances where the signal is unclear or fleeting—in a busy emergency room, for example—then individual differences in EI may be relevant.

### Limitations and future directions

These findings are limited in that they only pertain to static images, while real-world interactions are dynamic. Real-world interactions also have more context cues to emotions, such as body language [66] and tone of voice that may supplement the information lost by face coverings. Our findings generally align with findings of dynamic targets [31] and add to the understanding of how face coverings influence emotion perception, because these dynamic social interactions also result in more fleeting expressions with less processing time; however, future research should continue to investigate the effects that face masks have on the recognition of emotions in dynamic settings.

Future research may also further investigate how EI and other individual differences affect emotion recognition. The mixed findings in the current investigation and previous studies suggest EI as a predictor of emotion recognition is nuanced or contextual. Because of the sensitivity of this effect, the way EI is assessed might affect how it can be detected given the conditions of an experiment. In the current investigation we used a self-report measure of EI, as did Huc et al. [53], rather than an ability assessment, like the Mayer-Salovey-Caruso Emotional Intelligence Test [67] used by Carbon et al. [30].

Measuring EI with an ability assessment might be also effective at identifying individual differences in emotion processing under conditions of ambiguity.

## Conclusion

It is clear from our findings and others that face masks significantly impair emotion recognition and increase confusion rates. EI did not play a global role in emotion expression recognition, but higher EI resulted in higher accuracy when there were considerable constraints on viewing an expression. Confusion patterns are distinct for each emotion, with recognition of disgust most significantly impaired by face coverings and fear least impaired. Both AUs and the valance of emotions seem to influence confusion rates between emotions, where disgust and neutral are often confused for other emotional expressions.

Post-pandemic, masks are still prevalent in our world: in healthcare settings, religious settings, labor settings, and still in everyday life as disease prevention. In each of these settings, there is a sensitive social balance potentially at risk due to misunderstood emotional expressions. It is important that we continue to investigate factors that can moderate this, even in small ways, as we saw a glimpse of with EI in this study.

## Author contributions

**Conceptualization:** Andrea R. Libby.

**Formal analysis:** Collin Scarince.

**Investigation:** Andrea R. Libby.

**Methodology:** Andrea R. Libby, Collin Scarince.

**Supervision:** Collin Scarince.

**Writing – original draft:** Andrea R. Libby.

**Writing – review & editing:** Andrea R. Libby, Collin Scarince.

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
