## [Decision Letter · Decision Letter 0]

28 Mar 2025

Dear Dr. Scarince,

Thank you for submitting your manuscript to PLOS ONE. After careful consideration, we feel that it has merit but does not fully meet PLOS ONE’s publication criteria as it currently stands. Therefore, we invite you to submit a revised version of the manuscript that addresses the points raised during the review process.

The manuscript explores the impact of face masks on emotion recognition accuracy, considering factors such as emotional intelligence and stimulus timing. The topic explored is both relevant and engaging, and the study raises compelling questions; however, there are several significant issues that need to be addressed.

**Literature review and theoretical framing** requires substantial revision. I list some of the points that need to be taken into account:

The introduction is fragmented and lacks cohesion, presenting disconnected summaries of prior research without clearly linking them to hypotheses or study design.Key theoretical concepts, such as basic *versus* functional emotion theory, are conflated or inaccurately described.The review of existing literature is outdated and insufficient. Several recent and highly relevant studies—particularly those conducted post-COVID—were omitted, despite being directly related to the topic.

**Conceptual frameworks** need to be refined and clearly explained, and **terminology** needs to be reviewed and corrected:

Confusion exists between the concepts of *mediation* and *moderation* .Variables are inconsistently labeled (e.g., EI vs. SSEIT), causing confusion.Some statistical descriptions are vague or incorrect (e.g., unexplained statements like "the slope for the interaction is significant").

A clearer and more consistent presentation of the **methodology** and **data analysis** is needed to strengthen the manuscript.

Lack of justification for certain design choices, such as: *timing durations,*
*exclusion of neutral faces* , and *use of distractor items* .Poor explanation and interpretation of results; major findings are under-discussed or not contextualized.Inconsistent or unclear statistical reportingImportant variables such as *reaction times* are not analyzed or reported.

The **discussion** is limited, overly focused on selective sources (e.g., Carbon, 2020), and lacks depth. The manuscript makes causal claims without adequate statistical support, and certain interpretations—particularly those related to the confusion matrices—create a misleading impression of statistical testing where none was conducted. Certain **writing and formatting issues** require revision.

We look forward to receiving your revised manuscript.

Kind regards,

Livia Petrescu

Academic Editor

PLOS ONE

Journal Requirements:

2**.** Please include your full ethics statement in the ‘Methods’ section of your manuscript file. In your statement, please include the full name of the IRB or ethics committee who approved or waived your study, as well as whether or not you obtained informed written or verbal consent. If consent was waived for your study, please include this information in your statement as well.

3. We note that Figure 1 includes an image of a participant.

Reviewers' comments:

Reviewer's Responses to Questions

**Comments to the Author**

1. Is the manuscript technically sound, and do the data support the conclusions?

Reviewer #1: No

Reviewer #2: Yes

Reviewer #3: Yes

2. Has the statistical analysis been performed appropriately and rigorously?

Reviewer #1: No

Reviewer #2: Yes

Reviewer #3: Yes

3. Have the authors made all data underlying the findings in their manuscript fully available?

Reviewer #1: Yes

Reviewer #2: Yes

Reviewer #3: Yes

4. Is the manuscript presented in an intelligible fashion and written in standard English?

Reviewer #1: No

Reviewer #2: No

Reviewer #3: Yes

Reviewer #1: The article investigates whether wearing facial masks has effects on emotion recognition accuracy, and whether emotion intelligence and timing of the expression seen mediate the effects. The authors conducted two studies to exam the effects. The findings from the articles were interesting, and experiments were well designed. However, the writing, organizations and technical soundness needs to be significantly improved. I do not recommend the article to be accepted in its current form. I will discuss my comments below

1. The introduction is poorly written. This is my biggest concern with the article.

a. First of all, the discussion of emotion theory, facial mimicry, emotion intelligence, emotion expression do not connect with each other at all. All I see is a pile of short summaries of research papers, and they do not connect to the arguments. All the descriptions do not connect to your hypothesis

b. Discussion of emotion theory is not accurate. For example, page 3. Basic emotion theory states that there are some basic emotions that are universally recognized. They reflect individual’s internal states, originated by survival instincts and biological needs. The authors mixed the concept of basic emotion theory and functional emotional theory. This is not acceptable. Page 4., authors claims that mouth region relays more emotional information that eye region. This is also not accurate.

c. There are many more articles studying similar topics since covid-19, and the authors failed to recognize them. For example, when you are talking about facial regions, facial masks, and emotion recognition, especially when you cited Dimberg’s work on facial mimicry out of nowhere, this immediately reminded me of the work in Niedenthal lab (see Langbehn, A. T., Yermol, D. A., Zhao, F., Thorstenson, C. A., & Niedenthal, P. M. (2022). Wearing N95, surgical, and cloth face masks compromises the perception of emotion. Affective Science, 3(1), 105-117.) Yet, I did not find evidence of authors mentioning their group

2. Please make sure the statistical reporting, as well as overall article structure is up to APA guidelines and publisher-quality format.

a. The results reporting style is not consistent across two experiments.

b. Variables names are changing all the time. I have trouble following. What is the difference between sseit scores and EI. You are reporting both of them.

c. What do you mean by “the slope for the interaction is significant”?? These writings are not acceptable. You need to explain and interpret the results in a professional fashion.

3. In the results and analysis section, page 12 – first paragraph, why is the EI x time interaction a violation of assumption? To me, it’s the most interesting findings of the study. The writing does not make any sense

4. All results had no interpretations.

5. For experiment 1, I do not see why you are adding the timing as a factor? This was never discussed previously. And why did you choose 1 second in the experiment 2?

6. Discussion needs significant improvement.

a. Authors claimed causation between facial masks and emotion recognition accuracy. However, the analysis did not show any causality. Please be careful with the strong claim.

b. Again, a list of results reporting, yet no sounding discussion.

Reviewer #2: Summary:

The authors report the results of two experiments investigating the obfuscating effect of face masks on the recognition of facial expressions of emotion. They provide a conceptual replication and extension of prior work on masked emotion recognition by including distractor responses and manipulating the time participants had to view the photographs. Moreover, they examined the relationship between emotional intelligence and emotion recognition accuracy across viewing conditions (i.e., masked vs. fully visible and time constraints). Their studies add to a growing literature suggesting that face masks decrease emotion recognition accuracy.

Major issues:

• The authors discuss three early investigations of the effect of face masks on emotion recognition in their Introduction (Grundmann et al., 2021; Carbon, 2020; and Kastendieck et al., 2022). However, I find that these three studies provide an insufficient discussion of relevant literature to situate the current experiments in the body of work investigating the perception of masked facial expressions. Substantial work has been conducted since this data was collected that is directly relevant to their hypotheses. I suggest the authors conduct a more thorough overview of the literature in the Introduction to better situate this study in the broader literature.

o Emotional intelligence and recognition of masked facial expressions (the abstract states that no such work has been conducted). For example, Carbon et al. (2022) observed no relationships between self-report and behavioral measures of emotional intelligence and emotion recognition accuracy for masked or fully visible expressions of emotion. Likewise, no relationships were found between measures of emotional intelligence and participants confidence in their judgments. On the other hand, Huc et al. (in press) observed that emotional intelligence was positively related to emotion recognition accuracy for masked faces and especially for masked happy faces.

o Stimulus presentation timing. At least one study has examined stimulus presentation timing and emotion recognition judgments of masked and fully visible faces (Tsantani et al. 2022) which seems highly relevant to the authors’ work. Shepherd et al. (2022) also examined masked/fully visible facial expression recognition for stimuli shown for 250ms.

o Emotion recognition accuracy. The authors correctly report (p. 7 para. 2) that “Kastendieck et al. (2022) did not find a decrease in emotion recognition accuracy caused by face masks”. However, this null effect is compared to two studies that observed such an effect which gives the reader the impression that the evidence that face masks impair emotion recognition accuracy is more mixed than the current literature suggests (for example see McCrackin et al., 2023; Rinck et al., 2022; Wong et al., 2022; Gori et al., 2021; Ruba et al., 2020). Other studies (e.g., Langbehn et al., 2022) also specifically examined the impact of masks on facial expressions predominantly expressed on the upper and lower parts of the face which seems relevant to the authors’ discussion.

• In the Introduction, there is some conceptual confusion between mediation and moderation. The authors report (p. 7, para. 2) that participant age, participant sex, and target age have been found to mediate the relationship between face visibility (masked vs. fully visible) and emotion recognition accuracy. However, to my knowledge, the studies cited only support conclusions about moderation.

• The Introduction leads the reader to believe that the authors would predict a mask by EI or perhaps even a mask by EI by timing interaction (see the last sentence of p. 8, para. 2). Some clarification as to why the authors did not include these predictions would be helpful.

• More information about why EI is likely to be related to emotion expression recognition is also warranted in the Introduction. For one, results suggestions that high EI individuals show faster reaction times provides little evidence that they will also be more accurate at judging the emotion expressed. Moreover, additional interpretation of results like those of Prajapati (2021) would be helpful to show how they are related to accuracy.

• The confusion matrices included in Experiments 1 and 2 are appealing and provide insight into which emotion expressions were confused with which emotion labels. However, they are descriptive in nature. Therefore, the discussion of these patterns should be limited to descriptive language and terms like significant (e.g., p. 20, para. 1) which give the impression that statistical tests were conducted should be avoided. Alternatively, the authors might consider running such analyses to determine whether masks impair specific emotion expressions significantly more than others.

• The format and writing of the confusion patterns in the General Discussion is difficult to follow, for one, the authors report the percentage of errors here, but it was not immediately clear this was the case (previously they reported the proportion of responses in the tables).

o There also appears to be a typo here, “While unmasked fear was most often confused for disgust in the study of Carbon, and fear followed by disgust in both our experiments, in both our mask conditions fear was most often confused for first surprise then disgust.” This appears to say that fear was confused most often for fear in the authors’ experiments.

• The General Discussion would also benefit from reviewing additional literature.

o The authors focus their discussion on comparing their results to those of Carbon (2020), however, several additional studies have been conducted on the topic.

o Other areas also lack relevant citations (e.g., the authors speculate that religious face coverings may impair emotion recognition a prediction that has been tested [Fischer et al. 2011; Kret et al., 2012]).

Minor issues:

• In the Results section of Experiment 1 the authors note that a significant EI * time interaction was found for masked faces, however, they do not interpret the interaction until the Discussion. A brief interpretation of this finding would be helpful in the Results section.

• The introduction seems to indicate that the effect of facial coverings on emotion recognition accuracy and “the potential influence it has on social interactions” would be examined. However, the latter appears to be beyond the scope of the paper (see p.3, para. 1). Some clarification on what is meant by this would help focus the Introduction.

• There appears to be some discrepancy on the number of stimuli the participants in study 1 viewed. P. 10, para. 1 indicates that participants saw 60 experimental trials, however, p. 11, para. 2 indicates they saw 63.

• Some clarification is needed in the analysis strategy section for Experiment 2. Did the authors initially conduct an ANCOVA? If so, were all assumptions met?

• The last sentence of p. 28 appears to be incomplete.

• The manuscript would benefit from some additional editing to ensure clarity of the author’s message. For one, the introduction could be improved by including additional transitions and connections between the studies the authors discuss. A few examples of clerical errors are also added below.

o On p. 7 interpersonally is repeated twice: “Emotional intelligence (EI) was first described as a distinct intelligence by Salovey and Mayer [20] and is generally regarded as the ability to appropriately regulate and perceive emotions interpersonally and interpersonally [21].”

o On p. 17 the nationality of the participants is stated twice: “Participants were again all United States citizens and predominately White/Caucasian (71% White/Caucasian, 9% Black/African American, 9% Latino/Hispanic, 7% Asian, 2% Middle Eastern, 2% Multiracial) and citizens of the United States.”

o On p. 24, “These patterns changes…”

o On p. 25, Masked happiness still has high recognition likely because the one visible AU is only shared with anger and disgust, which are two opposite valences from happiness.”

o On p. 28, “Future research may also benefit to investigate further the investigation…” and “Consequently, these findings did not investigate the effect of either target race, participants race, in interactions between target and model race.”

References:

Carbon, C. C., Held, M. J., & Schütz, A. (2022). Reading emotions in faces with and without masks is relatively independent of extended exposure and individual difference variables. Frontiers in psychology, 13, 856971. https://doi.org/10.3389/fpsyg.2022.856971

Fischer, A. H., Gillebaart, M., Rotteveel, M., Becker, D., & Vliek, M. (2011). Veiled emotions: The effect of covered faces on emotion perception and attitudes. social psychological and Personality Science, 3(3), 266-273. https://doi.org/10.1177/1948550611418534 (Original work published 2012)

Gori, M., Schiatti, L., & Amadeo, M. B. (2021). Masking emotions: Face masks impair how we read emotions. Frontiers in psychology, 12, 669432. https://doi.org/10.3389/fpsyg.2021.669432

Huc, M., Bush, K., Berrigan, L., Cox, S., & Jaworska, N. (in press). The influence of emotional intelligence on facial expression processing in males and females with and without psychiatric illnesses. Personality and Individual Differences, 237, https://doi.org/10.1016/j.paid.2025.113040.

Kret, M., & De Gelder, B. (2012). Islamic headdress influences how emotion is recognized from the eyes. Frontiers in Psychology, 3, 23544. https://doi.org/10.3389/fpsyg.2012.00110

Langbehn, A.T., Yermol, D.A., Zhao, F. et al. Wearing N95, surgical, and cloth face masks compromises the perception of emotion. Affective Science 3, 105–117 (2022). https://doi.org/10.1007/s42761-021-00097-z

McCrackin, S. D., Capozzi, F., Mayrand, F., & Ristic, J. (2023). Face masks impair basic emotion recognition: Group effects and individual variability. Social Psychology, 54(1–2), 4–15. https://doi.org/10.1027/1864-9335/a000470

Pavlova, M.A., Moosavi, J., Carbon, CC. et al. Emotions behind a mask: The value of disgust. Schizophrenia, 9(58) (2023). https://doi.org/10.1038/s41537-023-00388-3

Rinck, M., Primbs, M.A., Verpaalen, I.A.M., & Bijlstra, G. (2022). Face masks impair facial emotion recognition and induce specific emotion confusions. Cognitive Research: Principles and Implications, 7(83). https://doi.org/10.1186/s41235-022-00430-5

Ruba AL, Pollak SD (2020) Children’s emotion inferences from masked faces: Implications for social interactions during COVID-19. Plos one 15(12): e0243708. https://doi.org/10.1371/journal.pone.0243708

Shepherd, J. L., & Rippon, D. (2022). The impact of briefly observing faces in opaque facial masks on emotion recognition and empathic concern. Quarterly Journal of Experimental Psychology, 76(2), 404-418. https://doi.org/10.1177/17470218221092590 (Original work published 2023)

Tsantani, M., Podgajecka, V., Gray, K. L. H., & Cook, R. (2022). How does the presence of a surgical face mask impair the perceived intensity of facial emotions?. PloS one, 17(1), e0262344. https://doi.org/10.1371/journal.pone.0262344

Wong, H.K. & Estudillo, A.J. (2022). Face masks affect emotion categorisation, age estimation, recognition, and gender classification from faces. Cognitive Research: Principles and Implications, 7(91). https://doi.org/10.1186/s41235-022-00438-x

Reviewer #3: The paper by Libby and Scarince describes two experiments investigating how emotion recognition accuracy is affected by face coverings when faces are presented for a limited viewing time, as well as how individual differences, such as emotional intelligence, modulate this effect. The authors found that recognition accuracy was lower for masked faces compared to unmasked faces in both experiments, with masked expressions of disgust and sadness showing the greatest increase in confusion. Furthermore, emotional intelligence influenced accuracy only when viewing time was particularly restricted, and parts of the face were covered.

Overall, the paper is well-written and clearly presented. I have only a few minor comments that may help improve the manuscript:

- Why were neutral face images not included as target stimuli? They could have served as a useful control condition. Please specify this point.

- It is unclear why the authors selected these specific viewing times, which are relatively long, instead of other durations. Did they base their choice on previous literature, or was there another criterion for selection? Please clarify.

- It is unclear why observation time was considered as a between-subjects factor. Could the authors provide a rationale for this choice?

- I was wondering why reaction times were not included as dependent variables. Is there a specific reason for this? Additionally, did the authors exclude trials with extremely long or short reaction times from the analysis?

- What was the rationale for introducing the four distractors within the response options? How were these specific distractors selected?

**Do you want your identity to be public for this peer review?** For information about this choice, including consent withdrawal, please see our Privacy Policy

Reviewer #1: No

Reviewer #2: No

Reviewer #3: **Yes: ** Andrea Ciricugno

---

## [Author Response · Author response to Decision Letter 1]

14 May 2025

Due to the substantial changes to the paper, we have summarized the major changes by section below and address specific actions to comments by the reviewers following the overall summary.

General Summary

Introduction (literature review and theoretical framing)

• We heavily restructured the introduction to focus on consistent finding from before and after the COVID-19 pandemic. We do still emphasis Carbon’s (2020 & 2022) work as the motivation and method for our experiments were heavily based on it, but we further emphasized how our work builds on previous findings.

• We clarified the use of basic emotional theory (and, subsequently, the facial action coding system in the discussion) as the main theoretical formulation for our experiments. To help emphasis this, we introduce and discuss action units in the introduction as the central aspect of how occlusion affects accuracy.

• Along with the restricting of the introduction, we incorporated more recent findings in our literature review. We incorporated more recent work related to face coverings post-pandemic and targeted the review of emotional intelligence to its role in interpreting emotional expressions.

Experiment 1 (method and results)

• We provided more clarity in the explanation for why we limited view time of the images as a way to increase the difficulty of the recognition task to investigate the effect of emotional intelligence.

• Based on feedback from Reviewer 2, we reanalyzed our results with target emotion as a within-subjects factor and included additional figures to accompany those results. We also provided clarification for using the separate regressions to investigate the interaction between the time targets were visible with participant emotional intelligence.

• To reduce repetition between the individual experiment results sections, discussion sections, and the general discussion, we combined the results are discussion for the individual experiments. This includes more interpretation along the presentation of the results.

Experiment 2 (method and results)

• We provided more explanation for the changes made in Experiment 2 based on the limitations of the method and findings of Experiment 1.

• We also implemented the same structural changes made to Experiment 1 in Experiment 2, including the additional factor of emotion expression in our analysis.

General Discussion

• As with the introduction, we also made substantial structural changes to the general discussion. Most of the discussion is still centered on interpreting recognition errors in the context of obscured and shared action units, but we relate more of our findings to other recent findings, particularly when discussing confusion rates.

• Additionally, we further discussed the mixed findings regarding emotional intelligence.

Reviewer 1

The article investigates whether wearing facial masks has effects on emotion recognition accuracy, and whether emotion intelligence and timing of the expression seen mediate the effects. The authors conducted two studies to exam the effects. The findings from the articles were interesting, and experiments were well designed. However, the writing, organizations and technical soundness needs to be significantly improved. I do not recommend the article to be accepted in its current form. I will discuss my comments below

1. The introduction is poorly written. This is my biggest concern with the article.

a. First of all, the discussion of emotion theory, facial mimicry, emotion intelligence, emotion expression do not connect with each other at all. All I see is a pile of short summaries of research papers, and they do not connect to the arguments. All the descriptions do not connect to your hypothesis

Action: We made substantial changes to the structure and content of the introduction. In terms of flow, we give a brief overview of the using action units to identify expressions and pre-pandemic basic research on the consequences of obscuring action units. We then review research on emotion recognition in response to the COVID-19 pandemic. This review is more comprehensive than our previous submission, but we do still emphasize Carbon’s work, as it was the primary basis for our set of experiments. We then briefly review research about individual differences in emotion recognition, including the possible role of emotional intelligence. This leads into the current investigation, where we are more explicit about the selection of our variables compared to the prior research we reviewed.

b. Discussion of emotion theory is not accurate. For example, page 3. Basic emotion theory states that there are some basic emotions that are universally recognized. They reflect individual’s internal states, originated by survival instincts and biological needs. The authors mixed the concept of basic emotion theory and functional emotional theory. This is not acceptable. Page 4., authors claims that mouth region relays more emotional information that eye region. This is also not accurate.

Action: Thank you for pointing that out. We have clarified our overview to stay within the scope of basic emotion theory. We also contextualized the conclusion from Blais et al. (2012) about the diagnostic value of the mouth region to be less definitive.

c. There are many more articles studying similar topics since covid-19, and the authors failed to recognize them. For example, when you are talking about facial regions, facial masks, and emotion recognition, especially when you cited Dimberg’s work on facial mimicry out of nowhere, this immediately reminded me of the work in Niedenthal lab (see Langbehn, A. T., Yermol, D. A., Zhao, F., Thorstenson, C. A., & Niedenthal, P. M. (2022). Wearing N95, surgical, and cloth face masks compromises the perception of emotion. Affective Science, 3(1), 105-117.) Yet, I did not find evidence of authors mentioning their group

Action: We broadened our literature review to include additional research on face coverings and emotion recognition, including the suggested reference.

2. Please make sure the statistical reporting, as well as overall article structure is up to APA guidelines and publisher-quality format.

a. The results reporting style is not consistent across two experiments.

Action: We adjusted our analysis to also include the target emotions in both experiments. Along with new approach, we have updated the reporting of the results of those tests and are reported consistently. The additional regression analysis in Experiment 1 is the primary difference between the two experiments and we provided additional information to interpret the results (e.g., simple slopes for interpreting the interaction and confidence intervals for the coefficients).

b. Variables names are changing all the time. I have trouble following. What is the difference between sseit scores and EI. You are reporting both of them.

Action: We changed to using EI except when specifically talking about the measure used throughout the paper.

c. What do you mean by “the slope for the interaction is significant”?? These writings are not acceptable. You need to explain and interpret the results in a professional fashion.

Action: We clarified this is in reference to interaction coefficient from the regression analysis.

3. In the results and analysis section, page 12 – first paragraph, why is the EI x time interaction a violation of assumption? To me, it’s the most interesting findings of the study. The writing does not make any sense

Action: We originally planned to analyze the effect of emotional intelligence as a covariate in an ANCOVA, under the assumption that those with higher emotional intelligence will generally have higher accuracy. That does assume that the influence (slope) of emotional intelligence is similar across conditions (homogeneity of slopes). We tested that assumption before conducting our planned ANCOVA and found that was not the case. We agree that this is an interesting finding, which is why we conducted the follow-up regressions. We included more interpretation of this interaction in the results section (inclusion of the simple slopes of each time condition) and discuss it further in the general discussion. We originally interpreted this finding cautiously given the marginal nature of the regression results.

4. All results had no interpretations.

Action: We originally tried to reduce some of the redundancy between the results and discussions of the individual experiments and then the general discussion. In the revision, we provide more interpretation of the results within each experiment.

5. For experiment 1, I do not see why you are adding the timing as a factor? This was never discussed previously. And why did you choose 1 second in the experiment 2?

Action: Because of the speed of processing facial emotions, we wanted to add some level of ambiguity (or time pressure) for any individual difference like EI to make a meaningful difference. If unlimited time is allowed to sample the stimulus, something as subtle as EI wouldn’t make a meaningful difference. We provided additional clarification leading up to and in Experiment 1.

6. Discussion needs significant improvement.

a. Authors claimed causation between facial masks and emotion recognition accuracy. However, the analysis did not show any causality. Please be careful with the strong claim.

Action: We made significant changes to the general discussion as well. We expanded our analyses in the experiments to test overall recognition accuracy across emotions. In addition, we framed our discussion of the confusion patterns as more descriptive of what we found and compare those patterns to other findings.

b. Again, a list of results reporting, yet no sounding discussion.

Action: Throughout the general discussion we compare our findings to other studies that used similar methods to better fit the current findings into prior research.

Reviewer 2

The authors report the results of two experiments investigating the obfuscating effect of face masks on the recognition of facial expressions of emotion. They provide a conceptual replication and extension of prior work on masked emotion recognition by including distractor responses and manipulating the time participants had to view the photographs. Moreover, they examined the relationship between emotional intelligence and emotion recognition accuracy across viewing conditions (i.e., masked vs. fully visible and time constraints). Their studies add to a growing literature suggesting that face masks decrease emotion recognition accuracy.

Major issues:

• The authors discuss three early investigations of the effect of face masks on emotion recognition in their Introduction (Grundmann et al., 2021; Carbon, 2020; and Kastendieck et al., 2022). However, I find that these three studies provide an insufficient discussion of relevant literature to situate the current experiments in the body of work investigating the perception of masked facial expressions. Substantial work has been conducted since this data was collected that is directly relevant to their hypotheses. I suggest the authors conduct a more thorough overview of the literature in the Introduction to better situate this study in the broader literature.

Action: We expanded our review of previous work to include work since 2022.

o Emotional intelligence and recognition of masked facial expressions (the abstract states that no such work has been conducted). For example, Carbon et al. (2022) observed no relationships between self-report and behavioral measures of emotional intelligence and emotion recognition accuracy for masked or fully visible expressions of emotion. Likewise, no relationships were found between measures of emotional intelligence and participants confidence in their judgments. On the other hand, Huc et al. (in press) observed that emotional intelligence was positively related to emotion recognition accuracy for masked faces and especially for masked happy faces.

Action: We have incorporated both of these studies into the manuscript (although we discuss Huc et al. in more depth in the discussion).

o Stimulus presentation timing. At least one study has examined stimulus presentation timing and emotion recognition judgments of masked and fully visible faces (Tsantani et al. 2022) which seems highly relevant to the authors’ work. Shepherd et al. (2022) also examined masked/fully visible facial expression recognition for stimuli shown for 250ms.

Action: We have included additional references to support the rationale behind limiting the view time of the images.

o Emotion recognition accuracy. The authors correctly report (p. 7 para. 2) that “Kastendieck et al. (2022) did not find a decrease in emotion recognition accuracy caused by face masks”. However, this null effect is compared to two studies that observed such an effect which gives the reader the impression that the evidence that face masks impair emotion recognition accuracy is more mixed than the current literature suggests (for example see McCrackin et al., 2023; Rinck et al., 2022; Wong et al., 2022; Gori et al., 2021; Ruba et al., 2020). Other studies (e.g., Langbehn et al., 2022) also specifically examined the impact of masks on facial expressions predominantly expressed on the upper and lower parts of the face which seems relevant to the authors’ discussion.

Action: We now provide a more comprehensive review of more recent research that does make it clear face coverings are consistently found to affect emotion recognition.

• In the Introduction, there is some conceptual confusion between mediation and moderation. The authors report (p. 7, para. 2) that participant age, participant sex, and target age have been found to mediate the relationship between face visibility (masked vs. fully visible) and emotion recognition accuracy. However, to my knowledge, the studies cited only support conclusions about moderation.

Action: Yes, we have clarified this in our revision.

• The Introduction leads the reader to believe that the authors would predict a mask by EI or perhaps even a mask by EI by timing interaction (see the last sentence of p. 8, para. 2). Some clarification as to why the authors did not include these predictions would be helpful.

Action: Leading into Experiment 1, we provided additional rational for the role of time. We initially expected a simple linear relationship between EI and accuracy across conditions because even the 1.5 s condition would be sufficiently difficult (obviously, not the case here).

• More information about why EI is likely to be related to emotion expression recognition is also warranted in the Introduction. For one, results suggestions that high EI individuals show faster reaction times provides little evidence that they will also be more accurate at judging the emotion expressed. Moreover, additional interpretation of results like those of Prajapati (2021) would be helpful to show how they are related to accuracy.

Action: Towards the end of the introduction, we further clarify that the time limitation was meant to leverage the possible processing speed effect of emotional intelligence.

• The confusion matrices included in Experiments 1 and 2 are appealing and provide insight into which emotion expressions were confused with which emotion labels. However, they are descriptive in nature. Therefore, the discussion of these patterns should be limited to descriptive language and terms like significant (e.g., p. 20, para. 1) which give the impression that statistical tests were conducted should be avoided. Alternatively, the authors might consider running such analyses to determine whether masks impair specific emotion expressions significantly more than others.

Action: We clarified the language about the confusion rates to be more descriptive of what we found here and tie it back to other studies that reported confusion matrices. We also reanalyzed our data by including the target emotion as a within-subjects factor. While this does not directly test specific frequencies of erroneous responses across each target expression, but it does allow us to compare correct response rates across emotions and make more comparisons to previous studies that used similar methods

---

## [Decision Letter · Decision Letter 1]

18 Jul 2025

Dear Dr. Scarince,

Thank you for submitting your manuscript to PLOS ONE. After careful consideration, we feel that it has merit but does not fully meet PLOS ONE’s publication criteria as it currently stands. Therefore, we invite you to submit a revised version of the manuscript that addresses the points raised during the review process.

We look forward to receiving your revised manuscript.

Kind regards,

Jerritta Selvaraj

Academic Editor

PLOS ONE

Journal Requirements:

Reviewers' comments:

Reviewer's Responses to Questions

**Comments to the Author**

Reviewer #2: (No Response)

Reviewer #3: All comments have been addressed

2. Is the manuscript technically sound, and do the data support the conclusions?

Reviewer #2: No

Reviewer #3: Yes

3. Has the statistical analysis been performed appropriately and rigorously?

Reviewer #2: Yes

Reviewer #3: Yes

4. Have the authors made all data underlying the findings in their manuscript fully available?

Reviewer #2: Yes

Reviewer #3: Yes

5. Is the manuscript presented in an intelligible fashion and written in standard English?

Reviewer #2: No

Reviewer #3: Yes

Reviewer #2: The current version of the manuscript represents a substantial improvement from the previous version. The authors have addressed many of the concerns noted in the initial review. There are, however, a few additional areas that warrant further revision before this manuscript should be accepted for publication.

1.The paragraph beginning on line 66 could benefit from additional editing to make the author's argument clearer.

1a. For example, in Lines 68-71 it is unclear what the “also” refers to.

1b. As reviewer 1 mentioned, Langbehn et al. (2021) is likely relevant here. They tested the prediction that the effect of face masks on emotion intensity judgments would be moderated by whether a given facial expression is predominately displayed using AUs from the upper versus lower part of the face.

2. There are at least two places in the Introduction where research is listed without a clear conceptual link to the current work. Moreover, this research is not reviewed in sufficient detail for the reader to understand what this previous work observed.

2a. Lines 84 – 85: Why are trustworthiness and interpersonal closeness judgments relevant to the current work? If they are relevant, the relevance needs explanation. Moreover, it is unclear whether masks increased, decreased, or had no effect on these judgments.

2b. Lines 105 – 107: The studies reviewed here also seem unrelated to the current work, and it is unclear whether these individual differences moderated the effect of masking on accuracy (or another outcome measure).

3. Lines 119 – 123:

3a. The null effect of self-reported EI on accuracy could potentially be explained by low internal reliability. However, the null relationship between the ability-based EI measure and accuracy could not be explained by this.

3b. The abstract also highlights the mixed findings between EI and emotion recognition accuracy. However, the introduction only reviews one study indicating that people with high EI respond faster when categorizing emotion expressions and one study that found no relationship between EI and recognition accuracy. Huc et al. (2025) is not mentioned until the Discussion, however, it seems highly relevant here.

4. The term “Lure” responses is used differently across the paper, which makes it confusing. Sometimes it is used to refer to distractor responses (those that are never the correct answer [e.g., Lines 334-337, 409-411]); it also appears to refer to all potential responses (i.e., Lines 307-309).

5. For Lines 309-312, what evidence do you have that the neutral response functioned as a decision default in Experiment 1? Participants rarely used the neutral response for unmasked targets but occasionally used it for masked targets. As written, this argument does not seem compelling.

6. The results of Experiment 2 do not seem to fully support the conclusion in lines 392-394. Experiment 2 did not find “increased confusion rates” for all emotion expressions (Figure 4). For example, anger showed no difference in accuracy, no change in confusion for happiness, and, if anything, reduced confusion with disgust and sadness when masked. Fear also showed a large decrease in confusion with disgust when masked compared to when unmasked.

7. The results of Experiment 1 do not seem to support the conclusions in Lines 395-396. Experiment 1 found a significant simple effect of EI on masked emotion recognition accuracy when targets were presented for 1 second. This was similar to the simple effect of EI when the targets were presented for 0.5 seconds.

8. Lines 426-428 would benefit from additional editing to make it clear which expressions are being confused with which expressions.

9. There is an error in Lines 436-437. This sentence currently says that AU 4 is a component of sadness, which was a common error for both sadness and disgust.

10. Lines 441-443 and Lines 452-453 appear to have conflicting citations. Proverbio and Cerri (2022) is cited as showing both a limited effect of masking on the recognition of anger and no effect (they found no effect). Leitner et al (2022) is also cited for both claims (they found no effect). The remaining citations in this paragraph need to be reviewed for accuracy.

11. Lines 484-486: Carbon (2022) did not find a relationship between EI (ability or self-reported) and emotion recognition accuracy. If the authors believe that Carbon’s results were found in error, they need to explain why this is the case. Moreover, Huc et al. (2025) found a positive relationship between self-reported EI and accuracy.

12. Huc et al. (2023) did not include EI in their study (Lines 456-457, 459-462). The correct reference is Huc et al. (2025).

a. Huc, M., Bush, K., Berrigan, L., Cox, S., Jaworska, N. (2025). The influence of emotional intelligence on facial expression processing in males and females with and without psychiatric illnesses. Personality and Individual Differences, 237, 113040. https://doi.org/10.1016/j.paid.2025.113040.

13. Typographic Errors

13a. Line 122: et al.

13b. Line 180: “was” should be “were”

13c. Line 352: missing “with”.

13d. Line 476: missing “and” before “still”.

Reviewer #3: (No Response)

**Do you want your identity to be public for this peer review?** For information about this choice, including consent withdrawal, please see our Privacy Policy

Reviewer #2: No

Reviewer #3: **Yes: ** Andrea Ciricugno

---

## [Author Response · Author response to Decision Letter 2]

21 Jul 2025

Response to Reviewers

We would like to begin by thanking Dr. Selvaraj, Dr. Ciricugno, and Reviewer 2 for their time and patience in reviewing our manuscript and providing additional feedback on our work.

Below are our responses to comments, point-by-point, along with the changes made in the manuscript.

Reviewer 2

The current version of the manuscript represents a substantial improvement from the previous version. The authors have addressed many of the concerns noted in the initial review. There are, however, a few additional areas that warrant further revision before this manuscript should be accepted for publication.

1.The paragraph beginning on line 66 could benefit from additional editing to make the author's argument clearer.

1a. For example, in Lines 68-71 it is unclear what the “also” refers to.

1b. As reviewer 1 mentioned, Langbehn et al. (2021) is likely relevant here. They tested the prediction that the effect of face masks on emotion intensity judgments would be moderated by whether a given facial expression is predominately displayed using AUs from the upper versus lower part of the face.

Action: We have revised this paragraph to clarify the overall reduction of accuracy when parts of the face are occluded and that some expressions are affected more than others. We also included reference to Langbehn when talking specifically about occluding the lower half of the face for specific emotions (reference 27).

Revision (ln 75): Given that distinct regions of the face are important for interpreting facial expressions, it is not surprising that expression recognition may be impaired if certain parts of the face are not visible [20-25]. While both the eye region and mouth region are significant informants of emotions, Kotsia et al. [17] and Blais et al. [16] determined that occluding the mouth region impairs overall emotion recognition more than occluding other facial regions. Accuracy is further reduced for expressions that have more key AUs in the mouth region than other facial regions, such as disgust [18,26,27]. These findings together indicate that limited visibility of the lower half of the face, such as when a face mask is worn, will impair overall emotion recognition by blocking the observer from necessary diagnostic information. These findings also indicate that face coverings will impair recognition of some emotional expressions more than others, based on the diagnostic information necessary for identification of that expression [27].

2. There are at least two places in the Introduction where research is listed without a clear conceptual link to the current work. Moreover, this research is not reviewed in sufficient detail for the reader to understand what this previous work observed.

2a. Lines 84 – 85: Why are trustworthiness and interpersonal closeness judgments relevant to the current work? If they are relevant, the relevance needs explanation. Moreover, it is unclear whether masks increased, decreased, or had no effect on these judgments.

2b. Lines 105 – 107: The studies reviewed here also seem unrelated to the current work, and it is unclear whether these individual differences moderated the effect of masking on accuracy (or another outcome measure).

Action: The inclusion of these references is to (1) illustrate the broader impacts of face coverings beyond less certainty of what expression someone is showing and (2) help make the connection to why emotional intelligence is relevant to investigate.

We decided to keep most of these references and have added additional context to help them fit in the respective sections of the paper.

Revision (ln 92): In addition, face masks impact recognition differentially based on the AUs covered by face masks. For example, emotions that are expressed more with the eyes, such as anger, are less affected by face masks than other expressions [38]. Research on broader effects on interpersonal perception have found wearing facial coverings lowers trustworthiness [33, 39] and reduce feelings of interpersonal closeness [39,40].

Revision (ln 117): Research on broader effects on social perception have found wearing facial coverings lowers feelings of trustworthiness toward the target [33, 39] and reduces feelings of interpersonal closeness [39,40].

Revision (ln 137): Individual differences in mental wellness have also been investigated. Men with high perceived stress have been found to take longer to identify masked happiness than men with low or moderate stress while the inverse was found for women [45].

3. Lines 119 – 123:

3a. The null effect of self-reported EI on accuracy could potentially be explained by low internal reliability. However, the null relationship between the ability-based EI measure and accuracy could not be explained by this.

3b. The abstract also highlights the mixed findings between EI and emotion recognition accuracy. However, the introduction only reviews one study indicating that people with high EI respond faster when categorizing emotion expressions and one study that found no relationship between EI and recognition accuracy. Huc et al. (2025) is not mentioned until the Discussion, however, it seems highly relevant here.

Action: We added Huc et al. (2025) to the introduction (and corrected the citation) to the following the summary of Carbon et al. (2022). We then contrasted the two methodologies to point out that the null effect Carbon reported could be due to the unrestricted time targets were visible, limiting the potential effect of EI.

Revision: Recently, Huc et al. [53] investigated of how several psychological individual differences—including perceived stress, loneliness, psychological disorders, and EI—affect expression recognition when faces are covered. Unlike Carbon’s [30] procedure, targets were presented for only 450 ms. Their results indicated that EI was a positive predictor for recognition accuracy, particularly for happy expressions. The difference in target presentation duration might account for the discrepancy between Huc et al.’s positive findings and Carbon’s null results. EI may only reliably predict expression recognition when there is high ambiguity. When an expression is unobstructed and visible for an extended time, both high- and low-EI individuals can effectively process the expression. The less clear the signal is, like when part of the face is covered and the expression is fleeting, the more individual differences in processing emotional signals might affect recognition.

4. The term “Lure” responses is used differently across the paper, which makes it confusing. Sometimes it is used to refer to distractor responses (those that are never the correct answer [e.g., Lines 334-337, 409-411]); it also appears to refer to all potential responses (i.e., Lines 307-309).

Action: "Lure" is meant to only include response options for expressions that were not in the image set.

Revision (ln 283): This question included ten response options including six basic emotions (anger, fear, surprise, sadness, disgust, and happiness) as well as four lures that were never the correct response (neutral, tiredness, annoyance, and excitement) [39].

Revision (ln 321): Second, we removed lure responses to the emotion recognition task, as some options in Experiment 1 could reasonably be conflated with one another based on subtle differences and interpretation of expressions.

5. For Lines 309-312, what evidence do you have that the neutral response functioned as a decision default in Experiment 1? Participants rarely used the neutral response for unmasked targets but occasionally used it for masked targets. As written, this argument does not seem compelling.

Action: This was based on the descriptive trends in the confusion matrix that lures were frequently selected in the masked condition and this was true for neutral responses for most emotions. We revised this paragraph to include another example of conflated expressions and changed the language around the neutral response to reflect our descriptive interpretation of how the lure responses affected participant responses.

Revision (ln 420): For instance, annoyance was frequently conflated with anger in both masked and unmasked conditions. Furthermore, neutral could be used as a functional “default” response when there are not strong signals for an expression given its higher response rate in the masked conditions for several target emotions.

6. The results of Experiment 2 do not seem to fully support the conclusion in lines 392-394. Experiment 2 did not find “increased confusion rates” for all emotion expressions (Figure 4). For example, anger showed no difference in accuracy, no change in confusion for happiness, and, if anything, reduced confusion with disgust and sadness when masked. Fear also showed a large decrease in confusion with disgust when masked compared to when unmasked.

Action: We adjusted language to specify this interpretation is regarding the higher diversity in error response in the masked condition.

Revision (ln 518): Regarding confusion rate patterns between mask conditions, we found more varied error responses across most emotions in the masked condition, although anger, fear, and happiness were more robust against confusion in general.

7. The results of Experiment 1 do not seem to support the conclusions in Lines 395-396. Experiment 1 found a significant simple effect of EI on masked emotion recognition accuracy when targets were presented for 1 second. This was similar to the simple effect of EI when the targets were presented for 0.5 seconds.

Action: We included additional speculation on the reasoning for not replicating the positive relationship for the 1 s view time from Experiment 1 in Experiment 2.

Revision (ln 523): In this experiment, we did not find a relationship between EI and recognition accuracy in general or for either mask condition. Although the results of Experiment 1 suggested a relationship between accuracy and EI would be found in the masked condition, the removal of the lure responses made the recognition task easier, thus reducing the effect of EI.

8. Lines 426-428 would benefit from additional editing to make it clear which expressions are being confused with which expressions.

Action: We provided clarification that the confusion patterns in that section are regarding happiness and surprise.

Revision (ln 617): Confusing happiness and surprise with a neutral expression [29,30] may be due to insufficient visible diagnostic information or lack of response confidence. The shared AUs between happiness and surprise also account for how frequently they are confused for each other [55].

9. There is an error in Lines 436-437. This sentence currently says that AU 4 is a component of sadness, which was a common error for both sadness and disgust.

Action: We corrected that the expression sadness and disgust were being compared to anger.

Revision (ln 628): AU 4 is also a component of anger, which was a common error for both sadness and disgust targets.

10. Lines 441-443 and Lines 452-453 appear to have conflicting citations. Proverbio and Cerri (2022) is cited as showing both a limited effect of masking on the recognition of anger and no effect (they found no effect). Leitner et al (2022) is also cited for both claims (they found no effect). The remaining citations in this paragraph need to be reviewed for accuracy.

Action: We reviewed the citations in this section (and across the paper). We separated the list of citations in this section to better align with the parts of the statement they support.

Revision (ln 633): The low recognition for fear is also consistent with previous studies that have found lower overall accuracy for fear [42, 59] and limited effects of masks [36, 42] or no effect [29-31, 60]. The AUs characteristic of anger around the eyes (AUs 4, 5, and 7) are all shared with at least two other emotions [19]. This may account for the relatively low accuracy in recognizing anger. The high rates of confusion with disgust may be attributed to the shared AUs 4 and 7, while the raised upper lip (AU 9) important for identifying disgust is obscured in masked conditions. Similarly, sadness shares AU 4 with anger, one of only two visible indicators of sadness when images are masked while AU 4 and 5 are shared between anger and fear, leaving only one of three visible indicators of fear. This suggests that confusion patterns for masked anger relate to lack of information for other emotions more than for lack of identifiers of anger. The inconsistent findings for anger between the experiments reported here reflect the inconsistency from previous studies. Some investigations have found effects of masks on recognition of anger [29, 31, 32, 42, 60] while others have found no difference [40, 41, 59]. Indeed, Mikayzaki et al. [35] also found mixed effects between the two experiments they conducted.

11. Lines 484-486: Carbon (2022) did not find a relationship between EI (ability or self-reported) and emotion recognition accuracy. If the authors believe that Carbon’s results were found in error, they need to explain why this is the case. Moreover, Huc et al. (2025) found a positive relationship between self-reported EI and accuracy.

Action: We were not necessarily trying to make the case that the results are in error, but that the effect of EI is contextual and sensitive to how it is defined and, thus, measured. We have clarified this in the final paragraph of our limitations and future directions section.

Revision: Because of the sensitivity of this effect, the way EI is assessed might affect how it can be detected given the conditions of an experiment. In the current investigation we used a self-report measure of EI, as did Huc et al. [53], rather than an ability assessment, like the Mayer-Salovey-Caruso Emotional Intelligence Test [67] used by Carbon et al. [30]. Measuring EI with an ability assessment might be also effective at identifying individual differences in emotion processing under conditions of ambiguity.

12. Huc et al. (2023) did not include EI in their study (Lines 456-457, 459-462). The correct reference is Huc et al. (2025).

a. Huc, M., Bush, K., Berrigan, L., Cox, S., Jaworska, N. (2025). The influence of emotional intelligence on facial expression processing in males and females with and without psychiatric illnesses. Personality and Individual Differences, 237, 113040. https://doi.org/10.1016/j.paid.2025.113040.

Action: We have included the correct reference.

13. Typographic Errors

13a. Line 122: et al.

13b. Line 180: “was” should be “were”

13c. Line 352: missing “with”.

13d. Line 476: missing “and” before “still”.

Action: We have revised the identified errors and made additional minor changes throughout the manuscript.

Academic Editor / Journal

Action: We have included the recommended references (see below).

Action: We included one reference that was erroneously excluded in the previously submission (Huc et al., 2025; Frontiers in Psychology).

We also removed one reference that is no longer needed following some of our revisions (Taylor & Asmundson, 2021; PLOS One).

No references have been retracted.

While revising your submission, please upload your figure files to the Preflight Analysis and Conversion Engine (PACE) digital diagnostic tool, https://pacev2.apexcovantage.com/. PACE helps e

---

## [Decision Letter · Decision Letter 2]

1 Aug 2025

The effect of face masks on confusion of emotional expressions

PONE-D-25-09712R2

Dear Dr. Scarince,

We’re pleased to inform you that your manuscript has been judged scientifically suitable for publication and will be formally accepted for publication once it meets all outstanding technical requirements.

Kind regards,

Jerritta Selvaraj

Academic Editor

PLOS ONE

Reviewers' comments:

Reviewer's Responses to Questions

**Comments to the Author**

Reviewer #2: All comments have been addressed

2. Is the manuscript technically sound, and do the data support the conclusions?

Reviewer #2: Yes

3. Has the statistical analysis been performed appropriately and rigorously?

Reviewer #2: Yes

4. Have the authors made all data underlying the findings in their manuscript fully available?

Reviewer #2: Yes

5. Is the manuscript presented in an intelligible fashion and written in standard English?

Reviewer #2: No

Reviewer #2: The authors have thoughtfully addressed all of my previous concerns with the manuscript. There are a few typographical issues (noted below) that I found upon further reviewing the manuscript. Note that the line numbers below are based on the current version of the manuscript without tracked changes.

1. There are a few inconsistencies in the reporting of the statistics.

a. Line 261: The Beta is reported here as .216 and .217 in Table 1.

b. p-values are sometimes reported with a leading zero (e.g., lines 235, 238, 244, 373, and 378). I recommend removing the leading zeroes to match the rest of the manuscript (e.g., lines 245, 247, 248, 266).

c. Betas and F-values are predominantly reported to 2 decimal places; however, there are a few places where they are reported to 3 decimal places (e.g., lines 235 and 261, and Table 1), and there is one that is only reported to one decimal place in line 371. I recommend keeping these consistent.

2. Line 54 appears to have an omitted word between “result” and “activation” or needs some revision.

3. There appears to be an extra space on line 141 before “To that end…”

**Do you want your identity to be public for this peer review?** For information about this choice, including consent withdrawal, please see our Privacy Policy

Reviewer #2: No

---

## [Editor Report · Acceptance letter]

PONE-D-25-09712R2

PLOS ONE

Dear Dr. Scarince,

I'm pleased to inform you that your manuscript has been deemed suitable for publication in PLOS ONE. Congratulations! Your manuscript is now being handed over to our production team.

Kind regards,

on behalf of

Dr. Jerritta Selvaraj

Academic Editor

PLOS ONE